# Two Clubroot-Resistance Genes, *Rcr3* and *Rcr9*^wa^, Mapped in *Brassica rapa* Using Bulk Segregant RNA Sequencing

**DOI:** 10.3390/ijms21145033

**Published:** 2020-07-16

**Authors:** Md. Masud Karim, Abdulsalam Dakouri, Yan Zhang, Qilin Chen, Gary Peng, Stephen E. Strelkov, Bruce D. Gossen, Fengqun Yu

**Affiliations:** 1Agriculture and Agri-Food Canada, Saskatoon Research and Development Centre, 107 Science Place, Saskatoon, SK S7N OX2, Canada; masud.karim@canada.ca (M.M.K.); abdulsalam.dakouri@canada.ca (A.D.); yan.zhang7@canada.ca (Y.Z.); qilin.chen@canada.ca (Q.C.); gary.peng@canada.ca (G.P.); bruce.gossen@canada.ca (B.D.G.); 2Department of Agricultural, Food and Nutritional Science, University of Alberta, Edmonton, AB T6G 2P5, Canada; stephen.strelkov@ualberta.ca

**Keywords:** *Brassica rapa*, *Plasmodiophora brassicae*, bulk segregant analysis, RNA sequencing, variant analysis, genetic mapping, clubroot resistance

## Abstract

Genetic resistance is widely used to manage clubroot (*Plasmodiophora brassicae*) in brassica crops, but new pathotypes have recently been identified on canola (*Brassica napus*) on the Canadian prairies. Resistance effective against both the most prevalent pathotype (3H, based on the Canadian Clubroot Differential system) and the new pathotypes is needed. BC_1_ plants of *Brassica rapa* from a cross of line 96-6990-2 (clubroot resistance originating from turnip cultivar ‘Waaslander’) and a susceptible doubled-haploid line, ACDC, exhibited a 1:1 segregation for resistance against pathotypes 3H and 5X. A resistance gene designated as *Rcr3* was mapped initially based on the percentage of polymorphic variants using bulked segregant RNA sequencing (BSR-Seq) and further mapped using Kompetitive Allele Specific PCR. DNA variants were identified by assembling short reads against a reference genome of *B. rapa*. *Rcr3* was mapped into chromosome A08. It was flanked by single nucleotide polymorphisms (SNP) markers (A90_A08_SNP_M12 and M16) between 10.00 and 10.23 Mb, in an interval of 231.6 Kb. There were 32 genes in the *Rcr3* interval. Three genes (*Bra020951*, *Bra020974*, and *Bra020979*) were annotated with disease resistance mechanisms, which are potential candidates for *Rcr3*. Another resistance gene, designated as *Rcr9^wa^,* for resistance to pathotype 5X was mapped, with the flanking markers (A90_A08_SNP_M28 and M79) between 10.85 and 11.17 Mb using the SNP sites identified through BSR-Seq for *Rcr3*. There were 44 genes in the *Rcr9^wa^* interval, three of which (*Bra020827, Bra020828, Bra020814*) were annotated as immune-system-process related genes, which are potential candidates for *Rcr9^wa^*.

## 1. Introduction

Clubroot, caused by *Plasmodiophora brassicae* Woronin, is a soil-borne disease of brassica crops that causes around 15% annual yield loss worldwide [1]. Integration of practices such as liming and early spring seeding can reduce clubroot levels, but is often not economical for large-scale canola production [2,3,4,5].

Clubroot is an emerging threat to canola production across the northern Great Plains of North America [6,7,8,9,10]. Identifying genetic resistance for use in resistant cultivars could be an effective strategy for clubroot management in canola [11,12]. Clubroot resistance (CR) is usually not durable [13], so monitoring to identify shifts in the virulence of the *P. brassicae* population is important [14].

The pathogen recognition systems that provide strong resistance in plants generally function in one of two ways. In one system, a pattern recognition receptor (PRR) in the plant interacts with a microbial/pathogen-associated molecular pattern (MAMP, PAMP) from the pathogen inside the apoplast, resulting in PAMP-triggered immunity (PTI). In the other system, a R gene encoded NBS-LRR (nucleotide-binding site leucine-rich repeat) protein in the plant interacts with an effector from the pathogen, usually inside the cytoplast, resulting in effector-triggered immunity (ETI) [15]. The NBS-LRR protein family can be subdivided according to their functional domain as toll/interleukin-1 receptor (TIR)-domain-containing (TIR-NBS-LRR protein) and coiled-coil (CC)-domain-containing (CC-NBS-LRR protein) subfamilies [16]. NBS-LRR related disease resistance is effective against obligate and hemibiotrophic pathogens [17].

*Brassica rapa* ssp. *rapifera* has been the sole source of resistance to clubroot in commercial cultivars of *B. rapa*, *B. napus*, and *B. oleracea* [18,19,20,21]. Across all *Brassica* spp., at least 15 CR loci from *B. rapa*, 22 quantitative trait loci (QTLs) from *B. oleracea*, and 16 QTLs from *B. napus* have been identified [21,22,23,24]. In *B. rapa*, chromosome A03 carried the highest number of loci, including *CRa* [25,26,27], *CRb* [28], *Crr3* [29,30], *CRk* [31], *PbBa3.1* and *PbBa3.3* [32], *Rcr1* [33], *Rcr2* [34], *Rcr4* [24], and *Rcr5* [35]. Another four chromosomes were identified as carrying CR loci: *Crr1* [36], *Rcr9* [24], and *CRs* [37] on A08; *Crr2* on A01 [38]; *Crr4* on A06 [38]; and *CRc* [31] and *Rcr8* on A02 [24]. *Crr1* was demonstrated to consist of two gene loci; the major locus, *Crr1a*, and the minor locus, *Crr1b* [39,40]. *Crr1a* is expressed as a TIR-NB-LRR disease resistance protein in the stele and roots, and was shown to have originated from the European fodder turnip ‘Siloga’ [40]. The CR genes *CRa* and *CRb^kato^* also encode TIR-NB-LRR class proteins [23,40,41,42]. Recently, several new sources of CR have been identified [11,43,44]. Additionally, resistance locus *CRs*, resistant to a Korean isolate of *P. brassicae*, was recently reported from turnip line SCNU-T2016 on chromosome A08 (10.7–11.5 Mb). Two genes, *Bra020918* and *Bra020876*, were reported as candidate genes for *CRs* [37].

Pathotype 5X was the first pathotype identified that overcame resistance in the first generation of CR canola cultivars in Canada [14], although many more resistance-breaking pathotypes have subsequently been identified [45]. Studies to identify sources of resistance to pathotype 5X identified two independent QTLs, *Rcr8* and *Rc9*, in the A genome on chromosomes A02 and A08 [24]. In addition, a single dominant gene, *Rcr7*, was identified that provided resistance to pathotypes 3H and 5X in the C genome on chromosome C07, which was homologous to a region on the A03 chromosome [46]. 

In several cases, a modifier gene has been reported to increase the efficacy of a dominant CR gene against a virulent pathotype. For example, *Crr1* alone provided resistance to a moderately aggressive field collection [36], but resistance against a more aggressive field collection required the presence of the modifier locus *Crr2* [38]. Enhancers and locus control regions are occasionally located away from candidate genes, with the result that communications may occur via histone modification or intra- and inter-chromosomal loops [47]. Such unusual relationships can complicate use of marker-assisted selection for CR genes.

Next-generation sequencing (NGS) technology is being used to accelerate conventional breeding [48,49]. For example, high-throughput, sequence-based, single-nucleotide polymorphism (SNP) markers have largely replaced PCR and hybridization-based markers in the last decade [50]. SNPs are the most frequent and the most abundant DNA sequence variant type in plant genomes, human genomes, and other organisms [51,52].

The advantage of RNA sequencing (RNA-Seq) over other NGS technologies is that RNA-Seq can be applied to both genetic mapping and expression analysis [53]. RNA-Seq also identifies higher numbers of functional variants from coding regions [54,55]. Coding sequences represent only 1–2% of the genome [56], making downstream data handling and analysis of RNA much easier than with DNA. Recently, RNA-Seq has been used to identify functional variants related to disease resistance [35,56,57]. Bulk-segregant analysis (BSA) identifies differences between two distinct characters. Using BSA together with RNA-Seq (BSR-Seq) provides an efficient approach for the identification of differential gene expression and a high-density of SNP markers for gene mapping and marker-assisted selection (MAS) [35,46].

A *B. rapa* line that was resistant to several pathotypes of *P. brassicae* but genetically distant from genotypes known to carry the major CR gene *Rcr1* was selected as the focus of the current study. The objectives of the study were to (1) map the CR gene(s) in the *B. rapa* line using BSR-Seq through analysis of percentage of polymorphic variants (PPV) described by Dakouri et al. [46], (2) use genome-wide DNA variant analysis to identify SNP markers for MAS, and (3) identify the most probable candidate gene(s) responsible for the resistance in the *B. rapa* line.

## 2. Results

### 2.1. Inheritance of CR in *B. rapa* Line 96-6990-2

The clubroot reaction of the parental lines, F_1_ and BC_1_ population were tested based on inoculation with each of two pathotypes of *P. brassicae*, representing pathotypes 3 or H (3H) and 5 or X (5X), respectively, as classified on the differentials of Williams’ (1966) or the Canadian Clubroot Differential Set [45] under controlled conditions using the ratings of 0–3 (Figure 1). Each plant of 96-6990-2 was resistant to pathotype 3H and each plant of ACDC was susceptible (Table 1). Each of the F_1_ plants (derived from reciprocal crosses) was also highly resistant. Segregation for resistant (R) and susceptible (S) reaction in the BC_1_ population was consistent with an expected ratio of 1:1 (Table 1). The same pattern of results in the parental lines, F_1_ and BC_1_ population was obtained with inoculation with pathotype 5X (Table 1). These results indicated that the resistance in 96-6990-2 to pathotype 3H was associated with a single dominant nuclear gene (designated as *Rcr3*) and 96-6990-2 was homozygous at the *Rcr3* locus. Similarly, resistance to pathotype 5X was also conferred by a single dominant gene (designated as *Rcr9^wa^*), which was homozygous in 96-6990-2.

### 2.2. RNA-Seq and Assembling Short Reads from BRS-Seq for Resistance to Pathotype 3H

In total, MiSeq produced 122.3 million (M) reads, with a mean of 20.4 M reads per biological replicate. From the bulked samples sequence, 50.2–55.4% reads in the R bulk with 107.5 M reads and 44.1–49.1% reads in the S bulk with 97.1 M were assembled into the chromosomes of the *B. rapa* ‘Chiifu’ reference genome v1.5 [58] (Appendix A, Table 2). Total sequence data yielded 18.95 Gb per biological replicate, with a mean yield of 6.32 Gb. Mean depth of coverage was 31.4-fold for the R bulk and 28.4-fold for the S bulk. A similar number of reads were assembled against the top and bottom strands of the reference genome using the pooled sample assembly method [57] (Table 2).

### 2.3. Mapping *Rcr3* for Resistance to Pathotype 3H Using PPV

In total, 654.1 K variants were identified in the R bulks and 637.7 K variants in the S bulks compared with the reference genome at Q-call ≥15 and depth ≥5 (Table 3). The R and S bulks contained 585.8 K and 571.0 K SNPs, and 68.3 K and 66.6 K InDels, respectively. The number of variants identified per chromosome in R and S bulk were positively correlated with chromosome length (*r* = 0.83 and 0.81).

The highest PPV (38.2%) and the lowest percentage of monomorphic variants (61.8%) were identified on chromosome A08 (Figure 2a); the range for the other chromosomes was 32.3–34.6% for polymorphic variants and 65.4–67.7% for monomorphic variants. Chromosome A08 carried the highest PPV relative to the other chromosomes. This indicated that *Rcr3* likely resided on A08. To locate the CR on A08, the distribution of PPV on the chromosome was analyzed. Two peaks of poly variants were observed; at 0–1 Mb (45.2%) and at 8–11 Mb (40.3–42.7%), providing two candidate genomic regions for *Rcr3* (Figure 2b).

### 2.4. Mapping of *Rcr3* Though Kompetitive Allele Specific PCR (KASP) Analysis

To define the genomic region of *Rcr3*, a total of 240 plants in the BC_1_ population were analyzed with three SNP sites from the 0–1 Mb and 11 SNP sites from the 8–11 Mb physical intervals of chromosome A08 through Kompetitive Allele Specific PCR (KASP) assay (Figure 3, Appendix A).

A linkage map was constructed based on the reaction of the 240 BC_1_ plants to pathotype 3H, together with their genotypic data from KASP analysis with the 14 SNP markers (Figure 3, Appendix A). *Rcr3* was flanked by SNP markers A90_A08_SNP_M12 (located at 9,997,211 bp of A08) and A90_A08_SNP_M16 (located at 10,228,875 bp of A08), which spanned an interval of 231.66 Kb (Figure 3). Therefore, *Rcr3* was located in the genomic region of 8–11 Mb as determined by analysis of PPV, but not in the 0–1 Mb region (Figure 3).

### 2.5. Gene Annotation in the *Rcr3* Target Region 

There are 32 genes in the *Rcr3* target region (10.00–10.23 Mb of chromosome A03) of the *B. rapa* reference genome v1.5. Blast2GO by a BLASTX search of *Arabidopsis thaliana* (Appendix A) was performed. Twenty-eight of these 32 genes were annotated. Three genes were associated with disease resistance mechanisms; *Bra020951* produces a protein from the glycosyl hydrolase family with a chitinase insertion domain-containing protein, *Bra020974* produces a protein from the leucine-rich receptor-like protein kinase family, and *Bra020979* produces a receptor-like protein 11.

### 2.6. Mapping *Rcr9^wa^* for Resistance to Pathotype 5X

A total of 108 BC_1_ plants were analyzed with four SNP markers flanking *Rcr3* (A90_A08_SNP_M11 and M12 above *Rcr3*, A90_A08_SNP_M22 and M28 below *Rcr3*) for mapping of the gene for resistance to pathotype 5X using KASP analysis (Figure 4, Appendix A). However, the gene responsible for resistance to the pathotype 5X was not in the interval, but below the marker A90_A08_SNP_M28. Three additional SNP markers were chosen to define the location of *Rcr9^wa^* (Figure 4, Appendix A). *Rcr9^wa^* was flanked by A08_SNP_M28 (located at 10,850,444 bp of chromosome A08) and A08_SNP_M79 (located at 11,173,147 bp of chromosome A08), covering an interval of 322.7 Kb. This represented a different interval from that of *Rcr3* (10.00–10.23 Mb) and *Crr1* gene *Bra020861* (10,809,433–10,825,238 bp) [40], but lay within the same interval as *Rcr9* (7.1–13.5Mb), a QTL to pathotype 5X previously mapped in *B. rapa* [24]. Therefore, the gene for resistance to pathotype 5X was designated as *Rcr9^wa^* since it was originated from turnip cultivar ‘Waaslander’.

The *Rcr9^wa^* region between the flanking markers included 44 genes, three of which (*Bra020827*, *Bra020828*, *Bra020814*) were annotated as immune-system-process related genes (Appendix A).

## 3. Discussion

NGS has ignited a revolution in life sciences. It allows the rapid development and application of genomics tools, especially DNA markers in plant genetics and breeding. BSA (genotyping bulks of plants with extreme phenotypes) and NGS have recently been applied to mapping and marker development for CR genes in *Brassica* spp. [34,35,46,57,59]. To minimize the complexity of plant genomes for data analysis and the cost for NGS, BSA has also been coupled with RNA-Seq for genetic mapping. In this study, we used an established BSR-Seq method to map *Rcr3* into chromosome A08 of *B. rapa*. A large number of SNP sites were obtained in the BSA-Seq project. The location of *Rcr3* was further defined and a second gene (*Rcr9^wa^*) was finely mapped using the SNP markers identified from BSA-Seq, demonstrating that BSA-Seq coupled with conventional linkage analysis is a powerful method to identify genes of interest.

In the current study, the reaction of BC_1_ [ACDC × (ACDC × 96-6990-2)] to pathotypes 3H and 5X was assessed. Pathotype 3H is prevalent on canola in Canada, while pathotype 5X is virulent on previously resistant canola cultivars [14].

Previous studies had identified and mapped *Rcr1*, *Rcr2*, and *Rcr4*, which produced resistance against multiple pathotypes in *B. rapa* lines including pak choy cv. ‘Flower Nabana’, Chinese cabbage cv. ‘Jazz’, and canola breeding line T19 [24,34,57]. In the current study, *B. rapa* line 96-6990-2, which is genetically distant from ‘Flower Nabana’, ‘Jazz’, and T19, was used as the resistance source (Appendix A). A 1:1 segregation ratio in the BC_1_ population in response to inoculation with either pathotype 3H or pathotype 5X indicated that resistance was controlled by a single dominant locus. *Rcr3* was associated with resistance to pathotype 3H and was mapped onto chromosome A08 using SNP markers developed through BSR-Seq. Additionally, *Rcr9^wa^* was associated with 5X and mapped onto chromosome A08. Leaf tissues were used in this study because collection was easier and less likely to carry microbial contaminants compare to root tissue. Additionally, the plants continued to grow well after leaf collection without producing any root damage. Clubroot is a root disease, but RNA-Seq data from root as well leaf tissue had previously been shown to produce good numbers of variants for CR gene mapping [34,35,46,57,59].

Previous reports indicated that all of the loci identified on chromosome A03 (*Rcr1*, *Rcr2*, *Rcr4, Rcr5*) conferred resistance to pathotype 3H [24,34,35,57]. In the current study, a new CR gene effective against 3H, named *Rcr3*, was identified on chromosome A08 between SNP markers A90_A08_SNP_M12 (9,997,211 bp) and A90_A08_SNP_M16 (10,228,875 bp) and spanning a physical interval of 231.6 Kb. The CR gene *Crr1a* had previously been mapped to chromosome A08 and cloned. It is highly homologous to gene *Bra020861*, which was located in the 10.81–10.83 Mb region of chromosome A08. Therefore, *Rcr3* is not allelic to *Crr1*. Similarly, *Crr1b* has been mapped to chromosome A08, about 712 Kb from *Crr1a* [39]. Although *Rcr3* and *Crr1b* share the same genomic region, *Crr1b* functions as a complementary allele for *Crr1a* and only has a minor role in CR [40]. The difference in role supports our conclusion that *Rcr3* and *Crr1b* represent different genes. *Rcr3* lies in the same genomic region as CR gene *CRs* [37], but the current study demonstrated that these two genes differ in pathotype specificity. This study supports a previous report that chromosome A08 of *B. rapa* carries a cluster of CR genes [39,44].

In the current study, *Rcr9^wa^* was fine mapped in a 322.7 Kb interval (10.85–11.17 Mb) within the *Rcr9* interval (7.1–13.5 Mb) on chromosome A08. *Rcr3* and *Rcr9^wa^* were mapped as being 1.17 Mb apart on chromosome A08, based on plant reaction to inoculation with pathotype 3H and 5X. The flanking SNP markers that had been used to locate *Rcr3* did not co-segregate with pathotype 5X resistance, so we conclude that two separate genes are responsible for resistance to pathotype 3H and 5X in the A genome of parental line 96-6990-2. A major locus, *Rcr7* on C07, was recently reported be associated with pathotype 3H and 5X resistance. However, common SNP markers co-segregated completely with the resistance; so, it could be a single dominant gene or tightly linked genes. The C genome *Rcr7* region is homologous to *B. rapa* chromosome A03 [46].

A total of 654.0 K variants were produced in the R bulks and 637.7 K variants in the S bulks. SNP variants were more abundant than InDels. The number of variants identified per chromosome was positively correlated with chromosome length (r = 0.83 and 0.81), as supported by a previous report [57].

Variant analysis across chromosomes has become the method of choice to identify quantitative and qualitative trait loci [60,61]. Variation of SNPs in the R and S bulks provided valuable information about the location of resistance genes on chromosomes. Polymorphic variants are a key factor for the identification of differences between two traits, and chromosomes with the highest PPV likely carry the gene(s) of interest [35,46,57]. In the current study, the highest numbers of polymorphic variants were found in chromosome A08, which indicated that *Rcr3* was located on A08. *Rcr3* was further mapped into the physical interval from 8 to 11 Mb of chromosome A08 though analysis of PPV and confirmed by the selected SNP sites in the interval through KASP analysis. However, a PPV peak from 0 to 1 Mb of chromosome A08 was also found, but it was irrelevant for *Rcr3*. One reason for this could be the low number of variants identified in the region due to a relatively low depth of RNA-Seq reads in the region [35], which causes a bias on the estimation of PPV.

*Rcr3* was mapped within a flanking region of 231.66 Kb from A90_A08_M12 (9,997,211 bp) to A90_A08_M16 (10,228,875 bp). Blast2GO analysis was conducted within the flanking region to identify the candidate genes corresponding to *Rcr3*. Three genes were annotated with disease resistance mechanisms that appear to be viable options as candidate genes for *Rcr3*; *Bra020951* produces a protein from the glycosyl hydrolase family with a chitinase insertion domain-containing protein, and *Bra020979* produces a receptor-like protein 11. *Bra020974* was annotated to produce a protein in the leucine-rich receptor-like protein kinase family. However, further clarification of the relationships among the genes or other candidates and *Rcr3* was beyond the scope of the present study.

Three genes (*Bra020827*, *Bra020828*, *Bra020814*) were annotated as immune-system-process related genes, but only one gene, *Bra020814* (11,173,076–11,174,598 bp), of the 44 genes identified in the region flanking *Rcr9^wa^* was associated with a disease resistance protein (TIR-NBS-LRR class) in *A. thaliana*. Each of these genes is a possible candidate gene for *Rcr9^wa^*. Our previous study identified a QTL (*Rcr9*) in *B. rapa* line T19 for resistance to pathotype 5X of *P. brassicae* through analysis of QTL [24]. *Rcr9* was roughly mapped in a large interval of chromosome A08. In another recent study, *B. napus* lines with *Rcr9* exhibited an intermediate level of resistance, while lines with *Rcr9^wa^* were highly resistant (unpublished data). Therefore, it is likely that *Rcr9^wa^* is a different CR gene from *Rcr9*. However, cloning of the genes is required to further clarify the relationship between *Rcr9* and *Rcr9^wa^*.

In summary, CR genes that provided resistance against pathotypes 3H (*Rcr3*) and 5X (*Rcr9^wa^*) were identified. Tightly linked SNP markers associated with *Rcr3* and *Rcr9^wa^* were identified, and will be valuable for use in marker-assisted selection for these genes in both *B. rapa* and *B. napus*. Searching for candidates for *Rcr3* and *Rcr9^wa^* in the respective mapping intervals was also described. PPV analysis of variants based on BSR-Seq was demonstrated to be an effective technique to identify genome-wide variants for KASP-based mapping.

## 4. Materials and Methods

### 4.1. Plant Materials for CR Genetic Mapping

A backcross mapping population was developed from a cross between two *B. rapa* lines, the clubroot-resistant line, 96-6990-2, and the susceptible double haploid line ACDC. The parental lines were provided by Dr. Kevin Falk at Saskatoon Research and Development Centre, Agriculture and Agri-Food Canada. The line 96-6990-2 was selected for use as the resistant parent [11] because it was a *B. rapa* canola breeding line that carried CR introgressed from a clubroot-resistant turnip cultivar, ‘Waaslander’, which was a derivation of ECD04 [62]. This source was selected because turnip belongs to a different genetic subgroup (Appendix A) from pak choy (*B. rapa* subsp. *chinensis*) cvs. ‘Flower Nabana’, ‘Jazz’, and breeding line T19, which were the sources of the resistance loci (*Rcr1*, *Rcr2*, and *Rcr4*), respectively, that had previously been identified on chromosome A03 with efficacy against pathotype 3H of *P. brassicae* [24,34,57]. In a preliminary screening, 96-6990-2 was resistant to both pathotype 3H and pathotype 5X (field isolates LG1, LG2, and LG3). In total, 240 BC_1_ plants were used to map resistance to pathotype 3H and 108 BC_1_ plants for resistance to pathotype 5X-LG2. Leaf samples from the 240 BC_1_ plants were used for BSR-Seq.

### 4.2. Plant Inoculation with Pathotypes 3H and 5X 

One field isolate of *P. brassicae* pathotype 3H (original pathotype 3, based on Williams’ differential set) and another of 5X (field isolate LG2) [14], both collected from canola fields in Alberta, Canada, were used in this study. To produce inoculum, seedlings of a susceptible cultivar were inoculated and maintained under controlled conditions. After 5–6 weeks, clubbed roots were harvested from infected plants and stored at −20 °C. After softening about 5 g of frozen club in a small amount of water, the material was homogenized in a blender for 2 min and strained through 2–3 layers of nylon mesh cloth. The resulting spore suspension was diluted with deionized water to produce a final concentration of 1 × 107 resting spores mL^−1^.

Seedlings were grown in tall, narrow plastic pots (164 mL Conetainers, Stuewe & Sons INC., Corvallis, OR, USA) filled with soil-less mix (Sunshine Mix 3, TerraLink Horticulture Inc., BC, Canada). The pots were filled with tightly packed, moist potting mix, and 2.5 mL of inoculum was added to soil three times; before sowing, after sowing, and at the stem base of each seedling at 7 days after sowing, to minimize disease escape.

The inoculated seedlings were maintained in a growth chamber set at 23 °C with frequent watering to maintain moist soil conditions. At 6 weeks after inoculation, each plant was rated for clubroot symptoms using a 0–3 scale [33], where 0 = no symptoms, 1 = a few small clubs, 2 = moderate clubbing, and 3 = severe clubbing (Figure 1). After the plants were rated for clubroot symptoms, the leaf samples that had been collected previously were separated into resistant (R) and susceptible (S) groups to create R and S bulks. Three biological replicates were assessed, with each biological replicate consisting of one R bulk and one S bulk. Each R bulk consisted of 30 plants with a clubroot severity rating of 0 and each S bulk consisted of 30 plants with a clubroot rating of 2 or 3. Plants with a rating of 1 were not included in either bulk because their reaction was considered intermediate.

In addition to the parental lines (ACDC—susceptible, 96-6990-2—resistant), two checks susceptible to one or two pathotypes were included in each assessment. One check was DH16516, a doubled-haploid line from single spore of *B. napus* cv. ‘Topas’, susceptible to both pathotypes 3H and 5X, and the other check was cv. ‘45H29′, a commercial canola cultivar of Pioneer Hi-Bred in Canada, resistant to pathotype 3H but susceptible to pathotype 5X.

### 4.3. RNA Extraction, Illumina Library Preparation, and Sequencing

At 2 weeks after inoculation, young, fresh leaves were harvested from each BC_1_ plant, immediately placed into liquid nitrogen, and stored at −80 °C until used. Total RNA was extracted from each bulk using an RNeasy^®^ Mini Kit (Qiagen, Toronto, ON, Canada). The total RNA concentration was measured with a Nanodrop2000 (Thermo Scientific™) and RNA quality was assessed using Bio-Rad ExperionTM (Bio-Rad Laboratories, Inc. USA) following the manufacturer’s instructions (Catalog #700-7103 #700-7105). The RNA quality for each bulk sample was deemed acceptable if the RNA quality indicator (RQI) value in the total RNA sample was ≥8.

A 100 ng aliquot of RNA from each bulk sample was used to prepare a cDNA library, following the manufacturer’s protocol (Illumina TruSeq^®^ RNASample Preparation v2 Guide, RS-122-9001DOC, Part # 15026495 Rev. F, March 2014, Illumina Inc., San Diego, CA, USA). The cDNA library concentration was measured with a Nanodrop2000 and cDNA quality with Bio-Rad ExperionTM. The cDNA samples were diluted and quantified using KAPA library quantification kits (KAPA Biosystem, Germany). Sample pooling was performed by mixing 10 nM l-1 cDNA from each R and S bulk replicate (e.g., R1 with S1). The final concentration (18–20 pM L^−1^) was used to prepare sequencing samples using a MiSeq^®^ reagent Kit V3 as per the manufacturer’s instructions (MS-102-3001, Illumina Inc.). RNA sequencing was performed with an Illuimna MiSeq^®^ System. All three biological replicates were sequenced twice to increase the depth and breadth of coverage.

### 4.4. Read Mapping, Variant Analysis, and Genetic Mapping by Analysis of PPV

Raw pair-end RNA sequencing reads were filtered to remove small fragments (≤50 bp), aligned to a *B. rapa* reference genome v1.5 (http://brassicadb.org/brad/downloadOverview.php), then assembled into 10 chromosomes using SeqMan NGen (DNASTAR.13 Lasergene Inc., Madison, WI, USA). Pooled sample assembly for variant analysis was conducted as described by Yu et al. [57] using SeqMan Pro and ArrayStar software in DNASTAR.13. A standard filter option (SNP% ≥ 15%, P not ref ≥ 90%, Q-call ≥ 15 and depth ≥ 5) was used to get consistent output. The PPV analysis [57], which was recently confirmed as a desirable approach for genetic mapping [46], was used for genetic mapping.

### 4.5. SNP Selection, KASP Analysis, and Linkage Analysis

High quality SNP loci (hetero type, non-synonymy and 40–60% SNP distribution in the R bulks and 0% SNP in the S bulks) were chosen for KASP analysis. The primers were designed by adding standard FAM or HEX reporter dye tails to the 3′ end of selected SNPs (Appendix A). The KASP assay was conducted using a StepOne Plus Real-Time PCR System (Applied Biosystems, Mississauga, ON, Canada). To assess marker association, genomic DNA was collected from each BC_1_ plant and from the parental lines. KASP markers were selected based on the RNA-Seq data from the first set of 240 BC_1_ plants (inoculated with pathotype 3H). The polymorphic SNP markers that had been identified were validated and used for mapping on a second set of 109 BC_1_ plants (inoculated with pathotype 5X-isolate LG2).

Linkage analysis and genetic map construction were conducted with JoinMap version 4.1 [63]. Genetic distances were converted from recombination frequencies using the Kosambi mapping function [64].

### 4.6. Identification of Potential Candidates for the Mapped Genes

The location of the SNP markers flanking *Rcr3* (A90_A08_SNP_M12 and A90_A08_SNP_M16) and *Rcr9^wa^* (A08_SNP_M28 and A08_SNP_M79) on the *B. rapa* reference genome v1.5 (http://brassicadb.org/brad/downloadOverview.php) was obtained using SeqMan Pro and ArrayStar components in DNASTAR.13 software. The information on the physical location, length, description, and gene ontology (GO) name for each gene in the flanking region of the reference genome was searched using Blast2GO [65]. The level of gene expression based on RPKM (Reads Per Kilobase of transcript per Million mapped reads) in the leaf tissue of the *Rcr3* interval from the RNA-Seq project was obtained using ArrayStar.

## Figures and Tables

**Figure 1 ijms-21-05033-f001:**
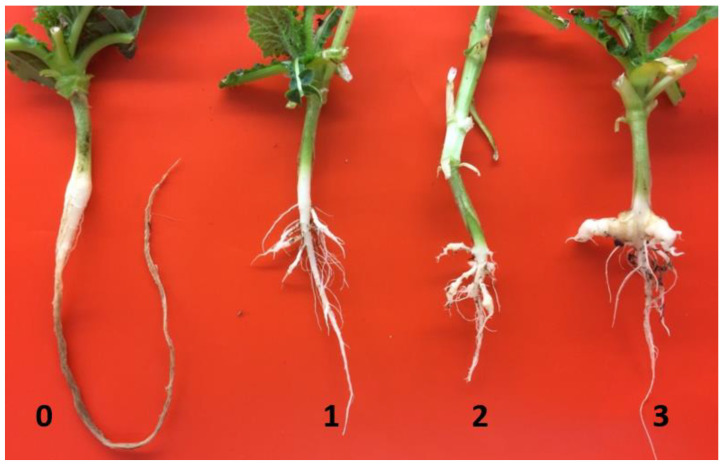
Clubroot reaction to pathotype 3H of the BC_1_ population derived from ACDC × (ACDC × 96-6990-2) at 6 weeks after inoculation; 0 = no clubs, 1 = a few small clubs, 2 = moderate clubbing, 3 = large clubs on the main and lateral roots.

**Figure 2 ijms-21-05033-f002:**
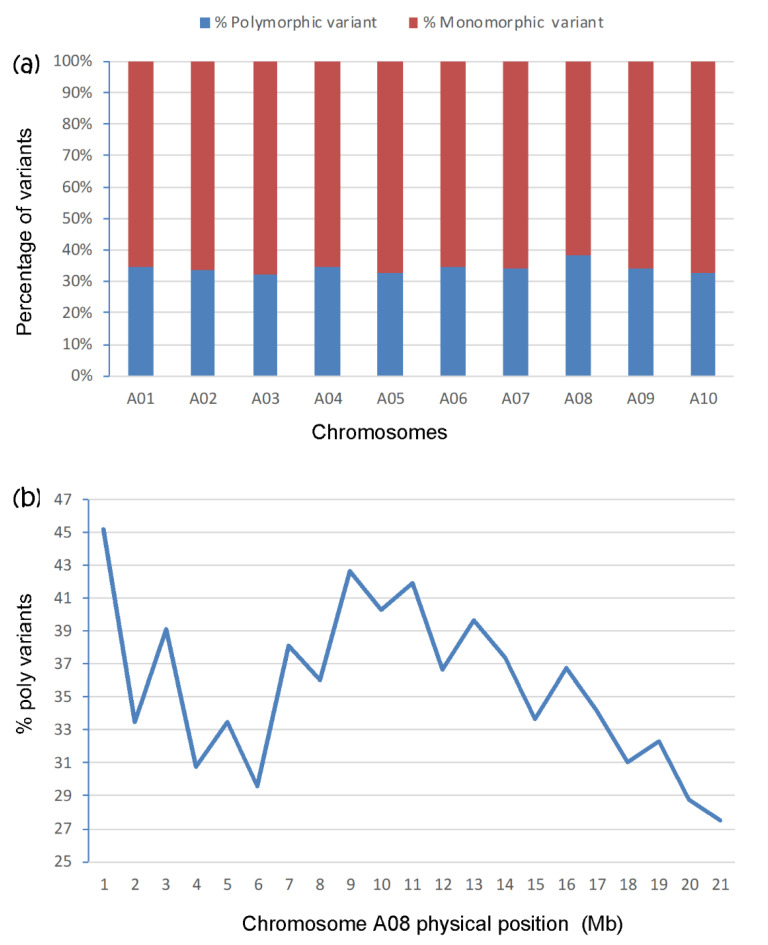
Analysis of bulked segregant RNA-Seq (BSR-Seq) to map *Rcr3* based on the reference genome of *Brassica rapa*: (**a**) the percentage (%) of monomorphic and polymorphic variants on each chromosome; (**b**) % polymorphic variants on chromosome A08.

**Figure 3 ijms-21-05033-f003:**
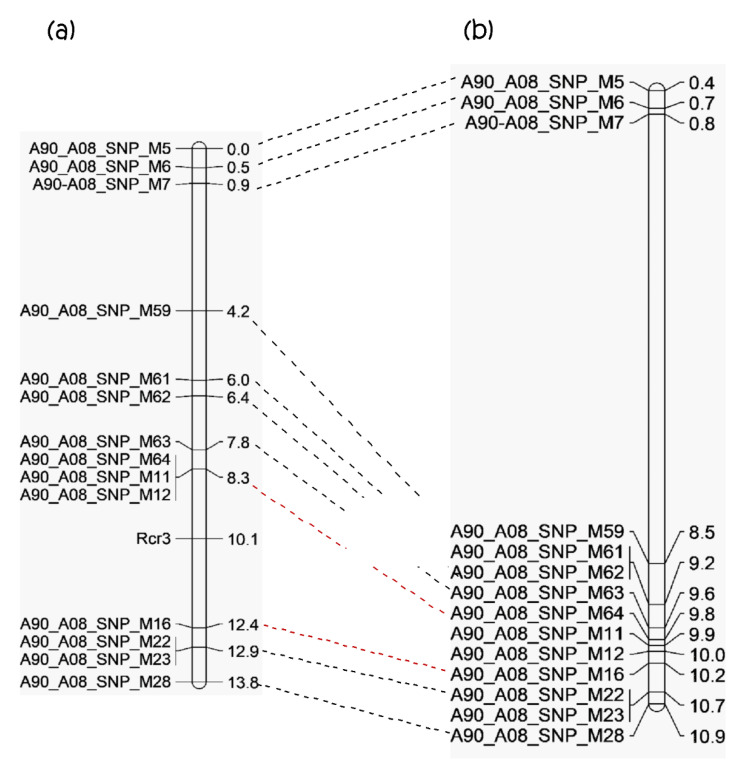
Genetic linkage map based on BC_1_ populations of *Brassica rapa* line 96-6990-2; (**a**) genetic map of the region in which the *Rcr3* gene is located (genetic distance on right), and (**b**) physical location of the *Rcr3* region (in bases, on right), with SNP markers connected with a broken line between the maps.

**Figure 4 ijms-21-05033-f004:**
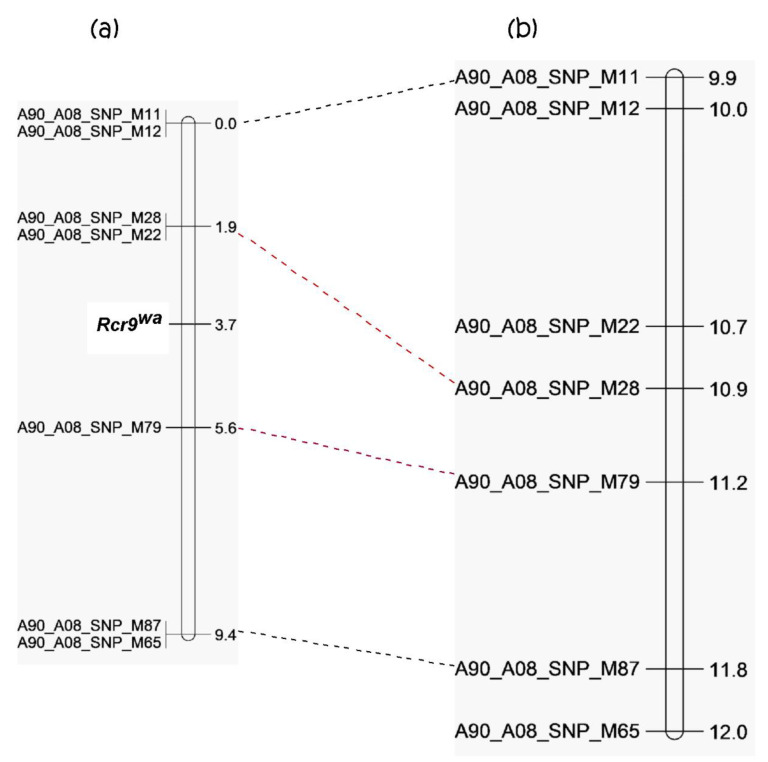
Genetic linkage map based on BC_1_ populations of *Brassica rapa* line 96-6990-2; (**a**) genetic map of the region in which the *Rcr9^wa^* gene is located (genetic distance on right), and (**b**) physical location of the *Rcr9^wa^* region (in bases, on right), with SNP markers connected with a broken line between the maps.

**Table 1 ijms-21-05033-t001:** Clubroot reaction (resistant, R; susceptible, S) to pathotypes 3H and 5X of *Plasmodiophora brassicae* in parental lines, F_1_, and backcross (BC_1_) progeny in crosses between *Brassica rapa* lines 96-6990-2 and ACDC, compared to an expected ratio of 1:1.

Parents and Crosses	Type	Clubroot Rating (0-3)					No. of Plants and Analysis			
		0	1	2	3	Total	R	S	χ2	P
**Pathotype 3H**										
96-6990-2	R parent	14	0	0	0	14	14	0	−	−
ACDC	S parent	0	0	0	14	14	0	14	−	−
ACDC × 96-6990-2	F_1_	7	0	0	0	7	7	0	−	−
96-6990-2 × ACDC	F_1_	7	0	0	0	7	7	0	−	−
ACDC × F_1_	BC_1_	118	0	0	122	240	118	122	0.06	0.81
**Pathotype 5X**										
96-6990-2	R parent	7	0	0	0	7	7	0	−	−
ACDC	S parent	0	0	0	2	7	0	7	−	−
ACDC × 96-6990-2	F_1_	14	0	0	0	14	14	0	−	−
96-6990-2 × ACDC	F_1_	14	0	0	0	14	14	0	−	−
ACDC × F_1_	BC_1_	77	0	0	60	137	77	60	2.10	0.15

A rating of 0 was defined as R, and ratings of 1–3 as S.

**Table 2 ijms-21-05033-t002:** Length and number of short read sequences assembled into chromosomes of the reference genome *Brassica rapa* ‘Chifu’ v1.5 in pooled sample assembly in the resistant (R) and susceptible (S) bulks of plants inoculated with pathotype 3H.

Chrom.	Ref. Genome Length (Mb)	Length of Accumulated Sequence (Mb)		Reads per Strand (×10^6^)					
				Top		Bottom		Total	
		R	S	R	S	R	S	R	S
A01	26.9	782.1	703.1	5.2	4.7	5.2	4.6	10.4	9.3
A02	27.0	676.3	607.9	4.6	4.1	4.4	4.0	9.0	8.1
A03	31.9	1127.5	1012.4	7.5	6.7	7.4	6.7	14.9	13.4
A04	19.3	601.4	548.9	4.1	3.7	3.9	3.6	8.0	7.3
A05	25.4	733.6	668.1	4.9	4.5	4.8	4.4	9.7	8.9
A06	25.3	912.2	799.2	6.1	5.3	6.0	5.3	12.1	10.6
A07	25.9	805.1	732.0	5.4	4.9	5.3	4.8	10.7	9.7
A08	20.9	724.6	669.3	4.8	4.5	4.8	4.4	9.6	8.9
A09	39.0	1186.2	1077.9	7.9	7.2	7.8	7.1	15.7	14.3
A10	16.4	563.2	513.5	3.8	3.4	3.7	3.4	7.5	6.8
Total	257.9	8112.1	7332.3	54.2	49.0	53.3	48.2	107.5	97.1

**Table 3 ijms-21-05033-t003:** Single nucleotide polymorphisms (SNPs), Insertions and deletions (InDels), and total variants identified in resistant (R) and susceptible (S) bulks relative to the *Brassica rapa* reference genome v1.5.

Chrom.	SNPs		InDels		Total	
	(×1000)		(×1000)		(×1000)	
	R	S	R	S	R	S
A01	57.5	56.1	6.1	6.0	63.6	62.0
A02	53.8	52.1	5.6	5.4	59.4	57.5
A03	87.1	86.5	9.1	9.0	96.2	95.5
A04	37.9	36.2	3.9	3.8	41.8	40.0
A05	58.0	57.0	6.2	6.1	64.2	63.1
A06	60.4	60.5	7.0	6.8	67.4	67.4
A07	58.2	56.4	6.0	5.9	64.2	62.3
A08	48.0	42.8	5.2	4.9	53.2	47.7
A09	80.4	79.4	14.4	14.2	94.8	93.5
A10	44.4	44.0	4.8	4.7	49.2	48.7
Total	585.8	571.0	68.3	66.6	654.1	637.7

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
