# Peer review of "Two Clubroot-Resistance Genes, Rcr3 and Rcr9wa, Mapped in Brassica rapa Using Bulk Segregant RNA Sequencing"

_ijms, 2020, doi:10.3390/ijms21145033_

Round 1
Reviewer 1 Report
The paper is interesting, well documented, the research methodology is appropriate, and the results contribute to the development of knowledge for the species Brassica rapa.
However, I do not understand why the research methodology is presented after the results, discussions. I think it should be inserted after the introductory part.
Author Response
Thank you so much for reviewing our manuscript. According to the Journal template, the research methodology was inserted after discussion, which is the requirement for IJMS.
Reviewer 2 Report
You have to add a small paragraph to the introduction about plant immunity explaining:pathogen-associated molecular pattern(PAMP), pattern recognition receptors (PRRs), mitogen-associated protein kinase (MAPK), effector-triggered immunity (ETI), NBS-LRR (nucleotide-binding site leucine-rich repeat) protein, (TIR-NBS-LRR protein), coiled-coil (CC) domains (CC-NBS-LRR protein), receptor-like protein (RLPs), transmembrane receptor-like kinases (RLKs), cytoplasmic kinases (CKs)
Author Response
Thank you so much for reviewing our manuscript. A paragraph was added between lines 45-54 to the introduction explaining plant immunity.
The pathogen recognition systems that provide strong resistance in plants generally function in one of two ways. In one system, a pattern recognition receptor (PRR) in the plant interacts with a microbial/pathogen-associated molecular pattern (MAMP, PAMP) from the pathogen inside the apoplast, resulting in PAMP-triggered immunity (PTI). In the other system, a R gene encoded NBS-LRR (nucleotide-binding site leucine-rich repeat) protein in the plant interacts with an effector from the pathogen, usually inside the cytoplast, resulting in effector-triggered immunity (ETI) [15]. The NBS-LRR protein family can be subdivided according to their functional domain as toll/interleukin-1 receptor (TIR)-domain-containing (TIR-NBS-LRR protein) and coiled-coil (CC)-domain-containing (CC-NBS-LRR protein) subfamilies [16]. NBS-LRR related disease resistance is effective against obligate and hemibiotrophic pathogens [17].
- Jones, J. D. G.; Dangl, J. L., The plant immune system. Nature 2006, 444, 323–329. doi: 10.1038/nature05286.
- Mchale, L.; Tan, X.; Koehl, P.; Michelmore, R. W., Plant NBS-LRR proteins: adaptable guards. Genome Biology 2006, 7, 212. doi:10.1186/gb-2006-7-4-212.
- Glazebrook, J. Contrasting mechanisms of defense against biotrophic and necrotrophic pathogens. Rev. Phytopathol. 2005, 43, 205–227 (2005).